health and disease and epidemiology/theoretical biology

projection, countermeasure, exit strategy, next-generation matrix, COVID-19, Japan

**Author for correspondence:**
Hiroshi Nishiura
e-mail: nishiura.hiroshi.5r@kyoto-u.ac.jp

†The first two authors contributed equally to this work.

# Projecting a second wave of COVID-19 in Japan with variable interventions in high-risk settings

Sung-mok Jung[1,2,†], Akira Endo[3,4,†], Ryo Kinoshita[1] and Hiroshi Nishiura[1,5]

[1]Kyoto University School of Public Health, Yoshidakonoe cho, Sakyo ku, Kyoto city 6068501, Japan
[2]Graduate School of Medicine, Hokkaido University, Kita 15 Jo Nishi 7 Chome, Kita-ku, Sapporo-shi, Hokkaido 060-8638, Japan
[3]Department of Infectious Disease Epidemiology, London School of Hygiene and Tropical Medicine, London, UK
[4]The Alan Turing Institute, London, UK
[5]CREST, Japan Science and Technology Agency, Saitama, Japan

S-mJ, 0000-0002-0787-4515; AE, 0000-0001-6377-7296; RK, 0000-0002-0116-4598; HN, 0000-0003-0941-8537

An initial set of interventions, including the closure of host and hostess clubs and voluntary limitation of non-household contact, probably greatly contributed to reducing the disease incidence of coronavirus disease (COVID-19) in Japan, but this approach must eventually be replaced by a more sustainable strategy. To characterize such a possible exit strategy from the restrictive guidelines, we quantified the next-generation matrix, accounting for high- and low-risk transmission settings. This matrix was used to project the future incidence in Tokyo and Osaka after the state of emergency is lifted, presenting multiple 'post-emergency' scenarios with different levels of restriction. The effective reproduction numbers ($R$) for the increasing phase, the transition phase and the state-of-emergency phase in the first wave of the disease were estimated as 1.78 (95% credible interval (CrI): 1.73–1.82), 0.74 (95% CrI: 0.71–0.78) and 0.63 (95% CrI: 0.61–0.65), respectively, in Tokyo and as 1.58 (95% CrI: 1.51–1.64), 1.20 (95% CrI: 1.15–1.25) and 0.48 (95% CrI: 0.44–0.51), respectively, in Osaka. Projections showed that a 50% decrease in the high-risk transmission is required to keep $R$ less than 1 in both locations—a level necessary to maintain control of the epidemic and minimize the risk of resurgence.

# 1. Introduction

Coronavirus disease 2019 (COVID-19) was first detected in Japan on 15 January 2020, only 8 days after the causative pathogen was identified as a novel coronavirus (severe acute respiratory syndrome coronavirus 2) [1,2]. A global pandemic was declared by the World Health Organization on 11 March 2020, as the epicentre of the outbreak expanded beyond Wuhan, China, to several other continents, including Europe. Beginning in late February 2020, Japan aimed to suppress transmission using the 'cluster-buster' strategy, which combines contact tracing with a call for the voluntary reduction of '3C' contacts (closed spaces, crowded places and close-contact settings) [3]. Despite an intensive effort, the number of untraced imported cases grew substantially, and local cases then began to shoot up in late March. In response, the government declared a state of emergency for seven prefectures (Tokyo, Saitama, Chiba, Kanagawa, Osaka, Hyogo and Fukuoka) on 7 April 2020, when the country reached a cumulative count of 3906 COVID-19 cases, excluding 672 cases diagnosed on the Diamond Princess cruise ship [4]; the state-of-emergency area was later expanded nation-wide (47 prefectures). The initial set of interventions included closing host and hostess clubs and asking for the voluntary limitation of non-household contact. Although the latter intervention (self-restraint-based reduction of contact) was not legally binding, the government requested that citizens reduce their physical contacts. Initially, the state of emergency was set to last until 6 May 2020, but this was later extended until 21 May 2020 for all prefectures except Hokkaido, Saitama, Chiba, Tokyo and Kanagawa. The state of emergency was lifted for all prefectures on 25 May 2020 [5].

In the absence of specific prevention or treatment measures, non-pharmaceutical interventions (NPIs)—especially physical distancing—play a key role in controlling the epidemic. Many countries have requested that their citizens reduce face-to-face contact; some countries have enforced additional stringent countermeasures, including closing non-essential businesses or even locking down cities [6,7]. In Japan, the legal enforcement of the state of emergency declaration was moderate, and the government relied on people's voluntary action and the resulting peer pressure to reduce contact at the community level. Soon after the declaration was issued, the incidence of COVID-19 (i.e. the number of new cases) was greatly reduced, easing the burden on healthcare facilities and public health centres. However, the broad restrictions on people's lives seen under the state of emergency could not be maintained for an extended period of time. Although the effects have not been explicitly quantified, the economic, social and psychological impacts have not been negligible. Given the temporal decline in the disease incidence, the next key questions are whether, when and to what extent we can maintain socioeconomic activities while sustaining successful suppression of the epidemic. Because it is unlikely that population-level immunity has been established, a full reopening without reservation may lead to a 'second wave' of the epidemic in Japan. To prevent such tragedy, a tailored reopening strategy that is well balanced between epidemic controls and people's socioeconomic activities needs to be considered.

Even after the easing of the current level of restriction, NPIs including physical distancing should remain the key part of the control strategy (at least until effective vaccines becomes widely available). However, the target of NPIs could be narrowed such that social functions are minimally affected. It has been suggested that the transmission of COVID-19 involves substantial individual-level variation (i.e. the majority of cases do not contribute to secondary transmission) and that the epidemic may be driven by 'superspreading' events [8]. Superspreading events frequently occur in specific settings (e.g. where meals are consumed or in closed environments) [9,10]. Many countries, even when lockdowns are not in place, have imposed restrictions on industries and activities they deem to be high risk [11]. Japan has requested the closure of host and hostess clubs and reduced opening hours for restaurants and bars (e.g. the Tokyo Metropolitan Government and local governments asked host and hostess clubs to close voluntarily and requested that restaurants and bars offer service no later than 20.00). If we focus intervention efforts on such high-risk settings to minimize the risk of superspreading events, we may be able to bring the epidemic under control while allowing other parts of society to gradually resume normal life. Such a political decision should be guided by quantitative assessment, comparing different targets and levels of interventions.

The present analysis aimed to assess possible scenarios for two of the most populous cities in Japan (i.e. Tokyo and Osaka), classifying the population into high- and low-risk settings. Using the reproduction numbers of COVID-19 derived from the estimated growth rates, we quantified the risk-based next-generation matrix and projected future epidemic scenarios with different levels of restriction.

# 2. Material and methods

## 2.1. Epidemiological data

We retrieved the daily incidence of confirmed COVID-19 cases in Tokyo and Osaka from 24 January (the earliest date of case reporting in Tokyo) to 26 May 2020. The dates of illness onset and laboratory confirmation were obtained from the Ministry of Health, Labour and Welfare [12] and from prefectural government websites [13,14]. The dataset includes a total of 6540 confirmed symptomatic COVID-19 cases. A total of 26 symptomatic cases with neither report of illness onset date nor laboratory confirmation date were excluded from the dataset.

## 2.2. Back-projection

Because there were 915 and 211 confirmed symptomatic cases with unknown illness onset dates in Tokyo and Osaka, respectively, we back-projected the onset date of these cases from the date of laboratory confirmation, using the empirical delay distribution between these dates. First, we fitted the right-truncated time delay from illness onset to laboratory confirmation, employing a Weibull distribution. We defined the likelihood as

$$L(\theta|S_k, O_k, T) = \prod_{k=1}^{K} \frac{h(S_k - O_k|\theta)}{H(T - O_k|\theta)}, \tag{2.1}$$

where $S_k$ is the date of laboratory confirmation of case $k$, $O_k$ is the date of illness onset of case $k$ and $T$ is the latest calendar date of observation (27 May 2020). The functions $h(\cdot)$ and $H(\cdot)$ are the probability density function and the cumulative distribution function of the Weibull distribution, respectively, and $\theta$ is the set of parameters (i.e. shape and scale parameters). The likelihood was maximized to determine the best-fit parameters to be used in the back-projection.

Second, using the parametrized delay distribution with the R package 'surveillance' [15], which is based on the non-parametric back-projection algorithm, all cases with unknown illness onset dates were back-projected. The back-projected cases by date of onset were then aggregated with cases with known dates of onset. We also accounted for right censoring with respect to the time delay from illness onset to laboratory confirmation. The 'nowcasted' number of new cases with the date of illness onset $t$ ($i(t)$) was calculated as

$$i(t) = \frac{i_{\text{reported}}(t)}{H(T - t)}, \tag{2.2}$$

where $i_{\text{reported}}(t)$ is the number of newly reported cases on date $t$. In the analysis, the nowcasted incidence for the latest 3 days was unreliable because the value of $H(\cdot)$ was too small; therefore, we excluded these dates (24 May 2020 onward) from the analysis.

## 2.3. Model and statistical analysis

We then conducted the projection analysis relying solely on the nowcasted date of illness onset. Our model consisted of three components: (i) exponential fitting, (ii) offspring distribution fitting, and (iii) next-generation matrix reconstruction. First, we fitted exponential curves to the observed case data to estimate the time-dependent reproduction numbers. We then constructed offspring distributions corresponding to the estimated reproduction numbers and approximated them by a mixture of two distributions for two types of host: low- and high-risk transmission settings. Finally, using the mean values for the two types of hosts, we reconstructed the next-generation matrix.

### 2.3.1. Estimation of growth rates

Let $i(t)$ be the incidence of infection (i.e. the number of new infections at time $t$). Given the constant next-generation matrix, we expect that $i(t)$ will converge to an exponential curve. We split the time axis into multiple periods (e.g. Period 1 $[t_0 < t < t_1]$, Period 2 $[t_1 < t < t_2]$), during which the next-generation matrix $K_n$ ($n = 1, 2, \ldots$) was assumed to stay constant, and we assumed that $i(t)$ in each period would swiftly converge to the asymptotic exponential curve such that $i(t)$ is well approximated by joint exponential curves

$$i(t) = i_{n(t)-1} \exp(r_{n(t)} t), \tag{2.3}$$

where $n(t)$ represents the period number to which $t$ belongs (i.e. $t_{n(t)} < t < t_{n(t)+1}$), and $r_n$ is the growth rate during Period $n$. The value of $i(t)$ at the beginning of Period $n$ is represented by $i_{n-1} = i(t_{n-1})$. Because we observe incidence using onset $j_t$ rather than infection $i(t)$ itself, the observation is delayed by the incubation period (whose relative frequency is represented by $f(\tau)$). The corresponding likelihood function is, therefore,

$$L(\mathbf{r}; \mathbf{j}) \;=\; \prod_t \text{Pois}(j_t; (i*f)(t)), \tag{2.4}$$

where $(i*f)(t) = \sum_{\tau=1}^{t} i(t-\tau)f(\tau)$ represents the expected incidence of illness onset. The values of the incubation period distribution $(f(\tau))$ were adopted from a previous study (mean = 5.6 days, standard deviation = 3.9 days) [16].

We used the following three time periods: (1) the early invasion period (before 26 March 2020 for Tokyo and before 18 March 2020 for Osaka), (2) the period after an alert was declared by the prefectural governor of each region (26 March–7 April 2020 for Tokyo and 19 March–7 April 2020 for Osaka), and (3) the period after the governmental declaration of a state of emergency (8 April–25 May 2020 for Tokyo and 8 April–21 May 2020 for Osaka). We fitted our model to the back-projected incidence by onset through 23 May 2020. The growth rates for the three periods ($r_1$, $r_2$ and $r_3$) were estimated by the Markov chain Monte Carlo (MCMC) method. Instead of directly sampling $r_1$, $r_2$ and $r_3$, we sampled $i(t)$ at the change points (i.e. $i_n$ [$n = 0, 1, 2, 3$]), the logarithms of which had improper flat priors ($\log(i_n) \approx \text{Unif}(-\infty, \infty)$). We ran 200 000 MCMC iterations to obtain 2000 thinned samples, and we discarded the first half as burn-in. We confirmed that the effective sample size of the resulting 1000 MCMC samples was larger than 500 for all parameters. The sampled $i_n$ was then translated to $r_n$ as

$$r_n = \left(\frac{i_n}{i_{n-1}}\right)^{-(t_n - t_{n-1})}. \tag{2.5}$$

We used R, v. 3.6.2 and v. 16.1.4 of the 'LaplacesDemon' package to implement the MCMC.

## 2.3.2. Approximation of the offspring distribution

Assuming that the generation time follows a gamma distribution with a mean ($T$) of 4.8 days and a coefficient of variation ($v$) of 0.5 days [17], we can obtain the converted value of the effective reproduction number from the growth rate using the Euler–Lotka equation [18,19]

$$R_n = (1 + r_n T v^2)^{1/v^2}. \tag{2.6}$$

The offspring distribution of COVID-19, the distribution of the number of secondary transmissions reproduced by a single primary case, has previously been modelled using a negative binomial distribution with the most plausible value of the overdispersion parameter $k \approx 0.1$ [8]. The mean of the negative binomial distribution, by definition, corresponds to the reproduction number. We assumed that the overdispersion parameter remains constant regardless of the reproduction number (i.e. the coefficient of variation is conserved) and that such a negative binomial distribution can be approximated as a mixture of two offspring distributions corresponding to low- and high-risk transmission subgroups

$$\hat{p}(x) = a p_L(x) + (1-a) p_H(x), \tag{2.7}$$

where $a$ is the mixture proportion (i.e. the proportion of the total secondary transmissions that are low-risk transmissions). Let $R_L$ and $R_H$ ($R_L < R_H$) represent the mean values of $p_L(x)$ and $p_H(x)$, respectively.

For each of the negative binomial offspring distributions corresponding to different reproduction numbers $R_n$ ($n = 1, 2$ and 3),

$$p_n(x) = \text{NegBin}(k = 0.1, \text{mean} = R_n), \tag{2.8}$$

we estimated the mixture distributions $\hat{p}(x)$ that provide the best approximation, where $p_L(x)$ and $p_H(x)$ are both assumed to be geometric distributions. We measured the deviation between $\hat{p}(x)$ and $p(x)$ by the Kullback–Leibler (KL) divergence (truncated at $x = 300$ for computational convenience),

$$D_{\text{KL}}(p(x) \| \hat{p}(x)) = \sum_{x=0}^{300} p(x) \log\left(\frac{p(x)}{\hat{p}(x)}\right), \tag{2.9}$$

and we minimized this value to find the best $\hat{p}(x)$. We obtained the two means, $R_L$ and $R_H$, for each MCMC sample for each period ($n = 1$, 2 and 3). We ensured that the minimized value of the KL divergence was less than 0.01. We used Julia-1.4.1 and the Optim.jl package for optimization, which were called from R with v. 0.17.1 of the 'JuliaCall' package.

### 2.3.3. Reconstruction of the next-generation matrix

From the mean values $R_L$ and $R_H$, corresponding to the low- and high-risk transmission subgroups, we reconstructed the next-generation matrix. The two-by-two next-generation matrix $K = (k_{ij})$ is expected to satisfy the following conditions: (i) the column sums are equal to $R_L$ and $R_H$, (ii) the eigenvalue of the matrix corresponds to $R_n$, and (iii) the off-diagonal entries of $K$ satisfy $qk_{LH} = (1 - q)k_{HL}$, where $q$ represents the relative size of contact in the high-risk subgroup (assuming that the risk of transmission is proportional to the contact rate). To ensure that the estimated next-generation matrix is positive-definite, we found that the parameter value of $q$ has to be within the range of 0–0.01 given the value of $a$, $R_L$ and $R_H$ in Period 1 (see electronic supplementary material, appendix S1). Thus, the value of $q$ was assumed to be 0.01 in the main analysis, and a sensitivity analysis was carried out adopting a smaller value of $q$ (0.001).

Let $R$ and $\rho$ be the two eigenvalues of $K$ (i.e. $R > \rho$) and $V = \begin{pmatrix} a & b \\ 1-a & 1-b \end{pmatrix}$ be the corresponding eigenvector matrix. We can then decompose $K$ as

$$K = \begin{pmatrix} a & b \\ 1-a & 1-b \end{pmatrix} \begin{pmatrix} R & 0 \\ 0 & \rho \end{pmatrix} \begin{pmatrix} a & b \\ 1-a & 1-b \end{pmatrix}^{-1}, \tag{2.10}$$

where $a$, $b$ and $\rho$ are unknowns to be determined by the binding conditions: $R_L = k_{LL} + k_{HL}$ and $R_H = k_{LH} + k_{HH}$. From condition (ii), we get

$$\left. \begin{array}{l} R_L = \dfrac{(1-b)R - (1-a)\rho}{a - b} \\[2mm] R_H = \dfrac{a\rho - bR}{a - b} \end{array} \right\} \tag{2.11}$$

and

When we fix $a$ as a constant, the above simultaneous equations for ($\rho$, $b$) have a solution if and only if $R = aR_L + (1 - a)R_H$. Suppose the value of $a$ satisfies this condition; such an $a$ corresponds to that in equation (2.7). Under such conditions, the two equations in (2.10) are reduced to a single equation

$$\rho = R_H - b(R_H - R_L). \tag{2.12}$$

In addition, to satisfy condition (iii), we require

$$(1 - q - a)b = (1 - a)(1 - q), \tag{2.13}$$

which only has a solution when $q \neq 1 - a$. We solved the simultaneous equations given by equations (2.12) and (2.13) and obtained $K$ from equation (2.10) for each MCMC sample.

## 2.4. Projection of newly reported cases

For the future projections, four different scenarios were considered: Scenario 1, going back to the pre-emergency period (i.e. assuming the next-generation matrix resumes, taking on values observed in Period 1), and Scenarios 2–4, with 10%, 30% and 50% decreases in high-risk transmission (i.e. assuming the normal low-risk transmission rate [$k_{LL}$ and $k_{HL}$] but a 10%, 30% or 50% decrease in the high-risk transmission rate [$k_{LH}$ and $k_{HH}$]). Thus, in the scenarios assuming a decrease in high-risk transmission, the low-risk transmission fully returns to the original level from 21 May 2020 and 25 May 2020 in Osaka and Tokyo, respectively, whereas the high-risk transmission is partially suppressed.

Using the growth rates in each scenario derived from the adjusted next-generation matrices, the number of newly infected cases at time $t$, $i(t)$, was projected for each city as an exponential curve, and the expected incidence by the date of illness onset $t$ was calculated as

$$E(j_t) = \sum_{\tau = 1}^{t} i(t - \tau)f(\tau).$$

The median and 95% credible intervals (CrIs) were produced from the MCMC samples.

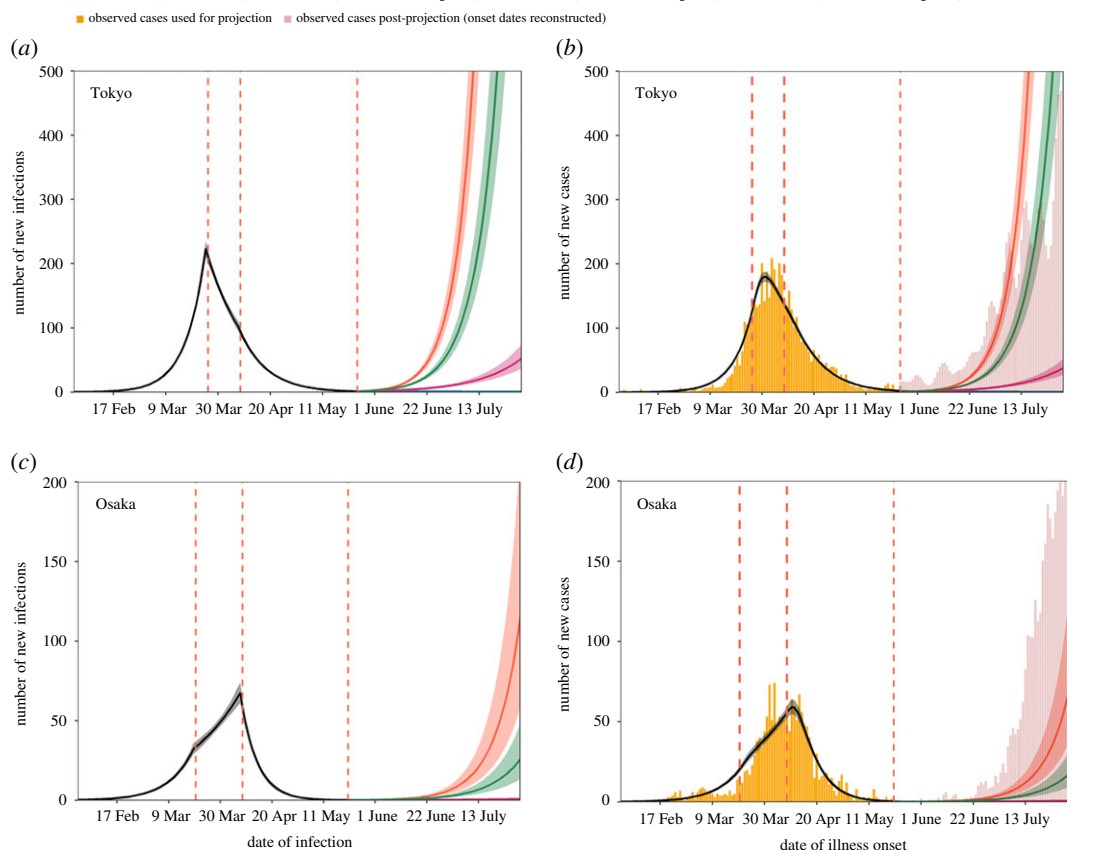

**Figure 1.** Estimated and projected trajectories of COVID-19 cases by date of infection and symptom onset including cases with unknown illness onset date in Tokyo and Osaka, Japan. (*a,c*) Estimated and projected numbers of COVID-19 cases by date of infection in Tokyo and Osaka, using all back-projected cases (yellow bars). The temporal trend in the number of new infections by 31 May 2020 was estimated using exponential curves with three different growth rates (black) and was then projected according to four scenarios (colours). The lines denote medians, and shaded areas represent 95% credible intervals (CrIs). Red dotted lines denote the dates when we assumed the trend changed (26 March, 8 April and 25 May 2020 for Tokyo; 20 March, 8 April and 21 May 2020 for Osaka). (*b,d*) Observed, estimated and projected numbers of COVID-19 cases by date of illness onset in Tokyo and Osaka. Yellow bars show the observed number of cases by the date of illness onset used for the projection, and red bars show the observed number of cases in the post-projection period (25 May–30 July 2020 for Tokyo; 21 May–30 July 2020 for Osaka). A small inconsistency was observed at the connecting part between the two time series because the post-prediction data lacked dates of illness onset, which were thus all reconstructed by the back-projection method from dates of laboratory confirmation. The estimated (black) and projected (colours) numbers of cases and their 95% CrIs are shown as lines and shaded areas, respectively.

In order to compare our projection with the actual post-projection incidence, the reported number of cases by the date of illness onset during the projection period (25 May–30 July 2020 for Tokyo; 21 May–30 July 2020 for Osaka) was back-projected from the date of laboratory confirmation. However, it should be noted that since the information of the date of illness onset of all cases in Tokyo was not available from June onwards, these post-projection data no longer represent exactly the same data source as the data used for projection, which may have caused a small discrepancy in incidence at the connecting point in the time series (figure 1).

## 3. Results

Table 1 presents the quantified next-generation matrix (*K*) for each time period in Tokyo and Osaka. The effective reproduction numbers for Tokyo were estimated as 1.78 (95% CrI: 1.73–1.82) in Period 1, 0.74 (95% CrI: 0.71–0.78) in Period 2 and 0.63 (95% CrI: 0.61–0.65) in Period 3. In Osaka, although the estimated reproduction number in Period 2 (i.e. after the request for self-restraint from the governor of Osaka) declined compared with that in Period 1 (1.58 [95% CrI: 1.51–1.64]), the estimated value in Period 2 was still above one (1.20 [95% CrI: 1.15–1.25]). In Period 3, the reproduction number was

**Table 1.** Estimated next-generation matrix of COVID-19 by time period in Tokyo Metropolis and Osaka, Japan. The values of $k_{ij}$ describe the transmission rates from group $j$ to group $i$, and $R_n$ is the reproduction number in Period $n$ derived from the estimated next-generation matrix. Upper and lower 95% credible intervals obtained from the MCMC samples are shown in parentheses.

| region | Period 1[a] | Period 2 | Period 3 |
|---|---|---|---|
| **including back-projected cases** | | | |
| Tokyo Metropolis | $k_{LL} = 0.02\ (0.02\text{–}0.02)$  $k_{LH} = 6.09\ (5.97\text{–}6.21)$  $k_{HL} = 0.06\ (0.06\text{–}0.06)$  $k_{HH} = 1.56\ (1.52\text{–}1.60)$  $R_1 = 1.78\ (1.73\text{–}1.82)$ | $k_{LL} = 0.03\ (0.03\text{–}0.03)$  $k_{LH} = 3.08\ (2.98\text{–}3.19)$  $k_{HL} = 0.03\ (0.03\text{–}0.03)$  $k_{HH} = 0.61\ (0.58\text{–}0.64)$  $R_2 = 0.74\ (0.71\text{–}0.78)$ | $k_{LL} = 0.03\ (0.03\text{–}0.03)$  $k_{LH} = 2.70\ (2.63\text{–}2.77)$  $k_{HL} = 0.03\ (0.02\text{–}0.03)$  $k_{HH} = 0.51\ (0.49\text{–}0.53)$  $R_3 = 0.63\ (0.61\text{–}0.65)$ |
| Osaka Prefecture | $k_{LL} = 0.02\ (0.02\text{–}0.02)$  $k_{LH} = 5.55\ (5.37\text{–}5.72)$  $k_{HL} = 0.06\ (0.05\text{–}0.06)$  $k_{HH} = 1.38\ (1.32\text{–}1.44)$  $R_1 = 1.58\ (1.51\text{–}1.64)$ | $k_{LL} = 0.02\ (0.02\text{–}0.03)$  $k_{LH} = 4.49\ (4.34\text{–}4.63)$  $k_{HL} = 0.05\ (0.04\text{–}0.05)$  $k_{HH} = 1.03\ (0.98\text{–}1.07)$  $R_2 = 1.20\ (1.15\text{–}1.25)$ | $k_{LL} = 0.03\ (0.02\text{–}0.03)$  $k_{LH} = 2.18\ (2.04\text{–}2.30)$  $k_{HL} = 0.02\ (0.02\text{–}0.02)$  $k_{HH} = 0.38\ (0.34\text{–}0.41)$  $R_3 = 0.48\ (0.44\text{–}0.51)$ |
| **excluding cases with unknown illness onset** | | | |
| Tokyo Metropolis | $k_{LL} = 0.02\ (0.02\text{–}0.02)$  $k_{LH} = 5.99\ (5.87\text{–}6.10)$  $k_{HL} = 0.06\ (0.06\text{–}0.06)$  $k_{HH} = 1.53\ (1.49\text{–}1.57)$  $R_1 = 1.74\ (1.69\text{–}1.78)$ | $k_{LL} = 0.03\ (0.03\text{–}0.03)$  $k_{LH} = 2.94\ (2.83\text{–}3.05)$  $k_{HL} = 0.03\ (0.03\text{–}0.03)$  $k_{HH} = 0.57\ (0.54\text{–}0.61)$  $R_2 = 0.70\ (0.67\text{–}0.73)$ | $k_{LL} = 0.03\ (0.03\text{–}0.03)$  $k_{LH} = 2.85\ (2.77\text{–}2.92)$  $k_{HL} = 0.03\ (0.03\text{–}0.03)$  $k_{HH} = 0.55\ (0.53\text{–}0.57)$  $R_3 = 0.67\ (0.65\text{–}0.70)$ |
| Osaka Prefecture | $k_{LL} = 0.02\ (0.02\text{–}0.02)$  $k_{LH} = 5.58\ (5.39\text{–}5.75)$  $k_{HL} = 0.06\ (0.05\text{–}0.06)$  $k_{HH} = 1.39\ (1.32\text{–}1.45)$  $R_1 = 1.59\ (1.52\text{–}1.65)$ | $k_{LL} = 0.03\ (0.03\text{–}0.03)$  $k_{LH} = 4.21\ (4.07\text{–}4.35)$  $k_{HL} = 0.04\ (0.04\text{–}0.04)$  $k_{HH} = 0.94\ (0.90\text{–}0.99)$  $R_2 = 1.10\ (1.06\text{–}1.15)$ | $k_{LL} = 0.03\ (0.03\text{–}0.03)$  $k_{LH} = 2.35\ (2.22\text{–}2.48)$  $k_{HL} = 0.02\ (0.03\text{–}0.02)$  $k_{HH} = 0.42\ (0.39\text{–}0.45)$  $R_3 = 0.53\ (0.49\text{–}0.56)$ |

[a]Three time periods were (1) the early invasion period (before 26 March 2020 for Tokyo and before 18 March 2020 for Osaka), (2) the period after an alert was declared by the prefectural governor of each region (26 March–7 April 2020 for Tokyo and 19 March–7 April 2020 for Osaka) and (3) the period after the governmental declaration of a state of emergency (8 April–25 May 2020 for Tokyo and 8 April–21 May 2020 for Osaka).

estimated at 0.48 (95% CrI: 0.44–0.51). In both regions, the transmission rate from low-risk subgroup to low-risk subgroup ($k_{LL}$) was almost constant over all three time periods, whereas the other three transmission rates ($k_{LH}$, $k_{HL}$ and $k_{HH}$) in Period 3 were less than half the values of these rates in Period 1 (table 1). There was little difference in the estimated next-generation matrices when cases with unknown dates of illness onset were excluded.

Figure 1 shows the projected number of new infections and cases using all back-projected cases in Tokyo and Osaka. In Tokyo, the projected number of newly infected cases in Scenario 1, assuming the same reproduction number as in Period 1 ($R_1 = 1.78$), surged again after the reopening on 26 May and reached the same value as the peak pre-reopening number of infected cases (i.e. 223 cases) on 4 July 2020. The estimated reproduction numbers in Scenarios 2 and 3 were 1.62 (95% CrI: 1.58–1.66) and 1.30 (95% CrI: 1.27–1.33), respectively, also suggesting resurgence. In Scenario 4 (assuming a 50% decrease in high-risk transmission), the estimated reproduction number was below the value of one (0.98 [95% CrI: 0.95–1.00]), and the projected epidemic was almost extinguished (figure 1a). Likewise, in Osaka, only Scenario 4 prevented a resurgence of cases; with all reported cases, the estimated reproduction numbers for Scenarios 2, 3 and 4 were 1.44 (95% CrI: 1.38–1.49), 1.16 (95% CrI: 1.11–1.20) and 0.87 (95% CrI: 0.84–0.90), respectively. The comparison between the estimated and observed numbers of new cases by date of illness onset in Tokyo and Osaka (figure 1b,d) suggested that the overall trends in the observed cases were well captured by our model in both regions. The reported number of cases during the projected period was consistent with Scenario 1 for Tokyo, while that in Osaka exceeded all scenarios in our projections.

Figure 2 shows the projected number of new infections and cases obtained from the dataset excluding cases with unknown illness onset. The estimated reproduction numbers exhibited similar trends to those produced with the back-projected dataset. The estimated reproduction numbers in Scenarios 2, 3 and 4 in Tokyo were 1.58 (95% CrI: 1.54–1.62), 1.27 (95% CrI: 1.24–1.30) and 0.96 (95% CrI: 0.93–0.98), respectively, whereas the reproduction numbers in Scenarios 2, 3 and 4 in Osaka were estimated as 1.45 (95% CrI: 1.38–1.50), 1.16 (95% CrI: 1.11–1.21) and 0.88 (95% CrI: 0.84–0.91), respectively.

In the sensitivity analysis using the smaller value of 0.001 as the relative size of the high-risk subgroup (as opposed to 0.01 in the main analysis), our conclusions did not substantively change. All entries related to the high-risk transmission rates ($k_{LH}$ and $k_{HH}$) showed very similar values. Although the values of $k_{HL}$ and $k_{LL}$ changed slightly, these differences had minimal impact on the results (electronic supplementary material, table S1). Additionally, considering the uncertainty of the overdispersion parameter ($k$), the sensitivity was assessed by varying this parameter in the range of 0.05–0.2 [7]. Decreasing values of $k$ (i.e. high dispersion) corresponded to larger estimated entries related to high-risk transmission rates ($k_{LH}$ and $k_{HH}$), which may indicate that decreasing high-risk transmission is particularly important in diminishing transmission when there is wide individual-level variation in the number of secondary transmissions (electronic supplementary material, tables S2 and S3). Throughout the sensitivity analyses, only the scenario with a 50% decrease in high-risk transmission was able to keep the reproduction number below one (see electronic supplementary material, appendix S2).

## 4. Discussion

The present study quantified the next-generation matrices classifying the population into two types, using the estimated reproduction number in three time periods. The parametrized system-generated projected numbers of new cases of COVID-19 in Tokyo and Osaka after lifting the state of emergency. Among the four scenarios, only the one with a 50% decrease in high-risk transmission (Scenario 4) was able to achieve containment of the epidemic (effective reproduction number $R_4 < 1$); the scenarios assuming smaller reductions resulted in a resurgence. In Tokyo, with 26 May 2020 as the reopening date, the projected number of new cases including all back-projected cases in early July 2020 was already as large as the peak (209 cases) observed on 3 April 2020 under the 'back to normal' scenario. Indeed, as of 9 August 2020, we have seen that the actual incidence in Tokyo exceeded 100 cases per day in early July. Although the numbers of new cases would be smaller than this 'normal' scenario under the 10%- and 30%-reduction scenarios, the number of cases is expected to grow under all three scenarios given that the reproduction number in each scenario is above one. In Osaka, the estimated reproduction number was also above one in each scenario except Scenario 4, with a 50% decrease in high-risk transmission. Thus, the projected number of cases showed a resurgence in Scenarios 1–3.

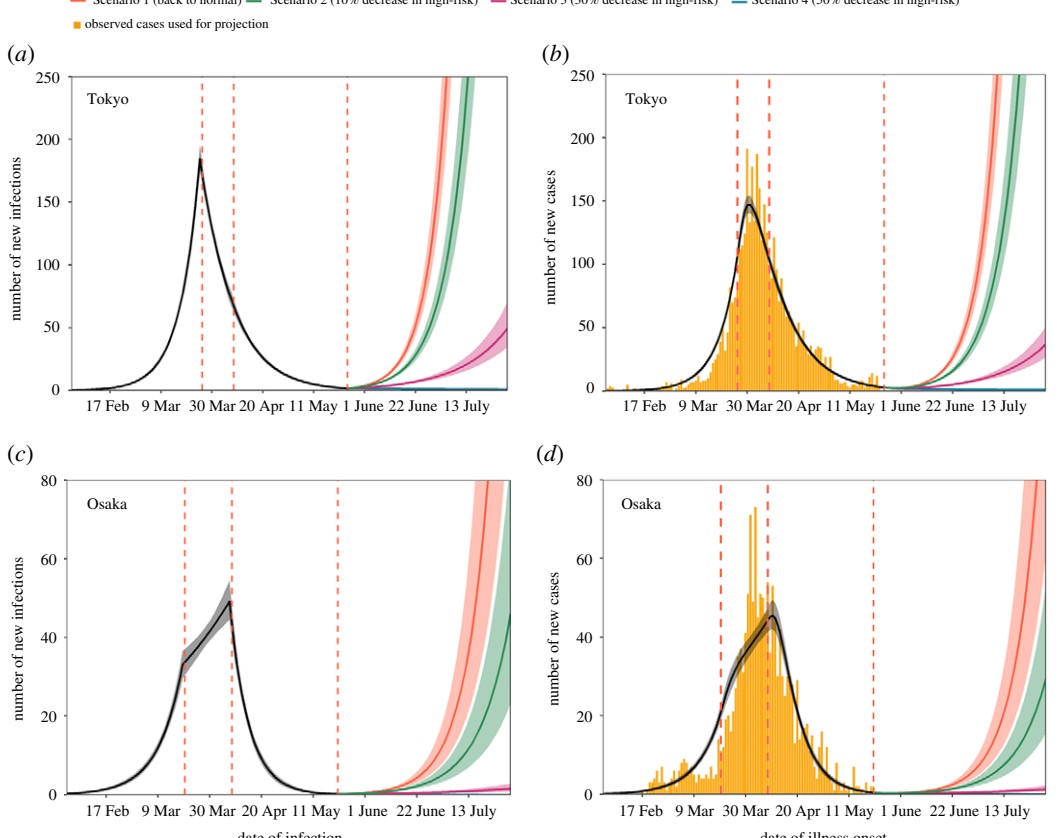

**Figure 2.** Estimated and projected trajectories of COVID-19 cases by date of infection and symptom onset, excluding cases with unknown illness onset in Tokyo and Osaka, Japan. (*a*,*c*) Estimated and projected numbers of COVID-19 cases by date of infection without back-projected cases in Tokyo and Osaka. The temporal trend in the number of new infections by 31 May 2020 was estimated using exponential curves with three different growth rates (black) and was then projected according to four scenarios (colours). The lines denote medians, and shaded areas represent 95% credible intervals (CrIs). Red dotted lines denote the dates when we assumed the trend changed (26 March, 8 April and 25 May 2020 for Tokyo; 20 March, 8 April and 21 May 2020 for Osaka). (*b*,*d*) Observed, estimated and projected numbers of COVID-19 cases by date of illness onset in Tokyo and Osaka. The bars show the observed number of cases by illness onset. The estimated (black) and projected (colours) numbers of cases and their 95% CrIs are shown as lines and shaded areas, respectively.

The number of observed COVID-19 cases in Tokyo was in line with the projected curve for the 'back to normal' scenario. As the state of emergency was lifted and economic activities started to resume since 25 May 2020, the newly reported cases from clusters in nightspots increased in Tokyo, which was likely to have resulted in the similar growth as Scenario 1. From around mid-July, the observed growth of cases slowed down compared to the projection, which coincided with the Tokyo governor's request for the residents to avoid high-risk settings and implementation of localized intensive PCR testing for nightspot industries in early July [20–23]. On the other hand, the observed post-projection incidence was higher than any of the projected scenarios in Osaka. The effective reproduction number estimated from the observed data was around 2 from mid-June to mid-July 2020 in Osaka [24], which was higher than the assumed baseline reproduction number $R_1 = 1.6$ (the value we estimated from the initial growth). These indicate that people's contact behaviour after the state of emergency was lifted in Osaka might have been above the level observed in Period 1. A possible explanation is that the awareness of people in Osaka may have already been raised in Period 1 because of four live-house-related clusters reported in the very early phase of the epidemic in Osaka [25].

Our main finding suggests that a decrease of more than 50% in high-risk transmission would be necessary to reduce the reproduction number of COVID-19 below unity. This is in line with the theory of type-reproduction number, $T$ [26], and the value of 50% is indeed consistent with $1 - 1/T$ for the high-risk group, which would be slightly more challenging than $1 - 1/R_1 = 1 - 1/1.7 = 41.2\%$. Until the development of effective vaccines, fully resuming all economic activity may lead to the second wave of COVID-19, and essential groups of hosts should thus be targeted for control to

sustain low levels of transmission. Although a temporary resurgence may be observed because of the stochasticity of transmission dynamics, the goal of a reproduction number of COVID-19 below one should be set for the exit strategy from restrictive guidelines because this would ensure the eventual extinction or at least continued suppression of the disease [27].

We approximated the overall offspring distribution, characterized as a negative binomial distribution, using a mixture of two geometric distributions with mean values of 0.08 and 7 for Period 1. The next-generation matrix reconstructed from these mean values was interpretable (i.e. positive values for all entries) only when the proportion at high risk ($q$) was very small ($q < 0.015$). This may indicate that the high-risk subgroup, generating seven secondary cases on average, would have to be disproportionately small in the population to be consistent with the observed reproduction number of 1.5. However, the proportion made up by the high-risk subgroup is larger in the infected population because of this population's relatively high risk of infection. In the approximated mixture offspring distribution, the high-risk subgroup accounted for 20% (according to the estimated value of $a$) of infectors and was responsible for 96% of the overall secondary transmissions. Although these are merely interpretations of the results of our mixture approximation and are not directly based on empirical data, our results suggest that a very small fraction of high-risk transmission could lead to a growing epidemic, as we observed in our projections. Thus, focusing on suppressing transmission in high-risk settings might have a greater impact than expected.

Three main limitations of this study should be discussed. First, our analyses relied only on reported case data. The reporting rate over time was assumed to be constant, which may not hold for the entire course of the current epidemic. Asymptomatic cases were also reported if they were identified by contact tracing; however, these cases were excluded from the analysis because of the uncertainty regarding the detection rate. Second, we assumed a constant value for the overdispersion parameter ($k = 0.1$) regardless of the reproduction number because this assumption ensures that the coefficient of variation of the negative binomial distribution is conserved; however, we have scarce data on how the variation in the offspring distribution of COVID-19 might change for different reproduction numbers. Third, we assumed that the negative binomial distribution could be well approximated by a mixture of two geometric distributions. We specifically selected the geometric distribution because it can efficiently capture the long tail of the negative binomial distribution, and the KL divergence of less than 0.01 suggests that our approximation was plausible.

# 5. Conclusion

The present study has quantitatively shown that a decrease of more than 50% in high-risk transmission could maintain a reproduction number of COVID-19 of less than one, keeping the epidemic under control for an extended period of time. Economic activities should be carefully resumed while implementing sufficient precautionary measures to suppress transmission in high-risk settings. Compared with stringent interventions such as lockdowns, our proposed exit strategy from restrictive guidelines, with the classification of high- and low-risk settings, allows socioeconomic activities to be maintained while minimizing the risk of a resurgence of the disease.

Ethics. The datasets used in our study were de-identified and fully anonymized in advance. The analysis of existing publicly available data without identity information did not require ethical approval.

Data accessibility. The time-series data on COVID-19 were publicly available on the websites of the Ministry of Health, Labour and Welfare [11] and the prefectures of Tokyo [12] and Osaka [13], and the incidence data by reporting date were available as a supplementary dataset. The relevant code for this research work is stored in GitHub: https://github.com/SungmokJung/Projection_Japan_COVID19 and have been archived within the Zenodo repository: https://doi.org/10.5281/zenodo.4576937 [28].

Authors' contributions. S.-m.J., A.E. and H.N. conceived the study and participated in the study design. All authors assisted in collecting the data. S.-m.J. and A.E. analysed the data, and S.-m.J., A.E., R.K. and H.N. drafted the manuscript. All authors have read and agreed to the published version of the manuscript.

Competing interests. A.E. received a research grant from Taisho Pharmaceutical Co., Ltd.

Funding. H.N. received funding from a Health and Labor Sciences Research Grant (grant nos. 19HA1003, 20CA2024 and 20HA2007); the Japan Agency for Medical Research and Development (AMED; JP19fk0108104 and JP20fk0108140); the Japan Society for the Promotion of Science (JSPS) KAKENHI (grant no. 17H04701), the Inamori Foundation, the GAP Fund Program of Kyoto University and the Japan Science and Technology Agency (JST) CREST program (grant no. JPMJCR1413). S.-m.J. received JSPS KAKENHI (20J2135800). A.E. is financially supported by the Nakajima Foundation and the Alan Turing Institute.

Acknowledgements. We would like to acknowledge the Local Health Centers and the Infectious Disease Surveillance sections of Tokyo Metropolitan Government and Osaka Prefecture for their work to make time-series data publicly available in real time. We thank Edanz Group (https://en-author-services.edanzgroup.com/ac) for editing a draft of this manuscript.

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
