## [Peer Review File · Royal Society Open Science]

Review History

Decision letter (RSOS-202169.R0)

Dear Dr Nishiura

On behalf of the Editors, we are pleased to inform you that your Manuscript RSOS-202169 "Projecting a second wave of COVID-19 in Japan with variable interventions in high-risk settings" has been accepted for publication in Royal Society Open Science subject to minor revision in accordance with the referees' reports. Please find the referees' comments along with any feedback from the Editors below my signature.

We invite you to respond to the comments and revise your manuscript. Below the Editors' comments (where applicable) we provide additional requirements. Final acceptance of your manuscript is dependent on these requirements being met. We provide guidance below to help you prepare your revision.

Please submit your revised manuscript and required files (see below) no later than 7 days from today's (ie 23-Feb-2021) date. Note: the ScholarOne system will 'lock' if submission of the revision

is attempted 7 or more days after the deadline. If you do not think you will be able to meet this deadline please contact the editorial office immediately.

on behalf of Dr Jianhong Wu (Associate Editor) and Glenn Webb (Subject Editor)
openscience@royalsociety.org

Associate Editor Comments to Author (Dr Jianhong Wu):

Associate Editor

Comments to the Author:

Japan has implemented a much stronger set of public health interventions and the projected period covers the current month when our decision for acceptance is being made. It is desirable for a follow-up remark to compare the current and the projection, and to explain the discrepancy if any.

===PREPARING YOUR MANUSCRIPT===

===PREPARING YOUR REVISION IN SCHOLARONE===

Author's Response to Decision Letter for (RSOS-202169.R0)

See Appendix A.

Decision letter (RSOS-202169.R1)

Dear Dr Nishiura,

It is a pleasure to accept your manuscript entitled "Projecting a second wave of COVID-19 in Japan with variable interventions in high-risk settings" in its current form for publication in Royal Society Open Science.

COVID-19 rapid publication process:

We are taking steps to expedite the publication of research relevant to the pandemic. If you wish, you can opt to have your paper published as soon as it is ready, rather than waiting for it to be published the scheduled Wednesday.

This means your paper will not be included in the weekly media round-up which the Society sends to journalists ahead of publication. However, it will still appear in the COVID-19 Publishing Collection which journalists will be directed to each week (<https://royalsocietypublishing.org/topic/special-collections/novel-coronavirus-outbreak>).

If you wish to have your paper considered for immediate publication, or to discuss further, please notify openscience_proofs@royalsociety.org and press@royalsociety.org when you respond to this email.

Please ensure that you send to the editorial office an editable version of your accepted manuscript, and individual files for each figure and table included in your manuscript. You can send these in a zip folder if more convenient. Failure to provide these files may delay the

processing of your proof. You may disregard this request if you have already provided these files to the editorial office.

You can expect to receive a proof of your article in the near future. Please contact the editorial office (openscience@royalsociety.org) and the production office (openscience_proofs@royalsociety.org) to let us know if you are likely to be away from e-mail contact – if you are going to be away, please nominate a co-author (if available) to manage the proofing process, and ensure they are copied into your email to the journal.

on behalf of Dr Jianhong Wu (Associate Editor) and Glenn Webb (Subject Editor)
openscience@royalsociety.org

Associate Editor Comments to Author (Dr Jianhong Wu):
Associate Editor
Comments to the Author:
Congratulate for a solid piece of important contribution to COVID-19 research.

Appendix A

Re: Projecting a second wave of COVID-19 in Japan with variable interventions in high-risk settings (RSOS-202169)

[Response to associate editor]

Comments to the Author:

Japan has implemented a much stronger set of public health interventions and the projected period covers the current month when our decision for acceptance is being made. It is desirable for a follow-up remark to compare the current and the projection, and to explain the discrepancy if any.

>>

Herewith, we wish to submit the revised manuscript by Sung-mok Jung, Akira Endo, Ryo Kinoshita and Hiroshi Nishiura for consideration for publication as an Original Article in International Journal of the Royal Society Open Science.

We have revised the manuscript in accordance with the editorial comment. Follow-up and discrepancy are described in Discussion section (P7, para 2).